# An unconstrained approach to systematic structural and energetic screening of materials interfaces

Giovanni Di Liberto [1] ✉, Ángel Morales-García [2] & Stefan T. Bromley [2,3] ✉

From grain boundaries and heterojunctions to manipulating 2D materials, solid-solid interfaces play a key role in many technological applications. Understanding and predicting properties of these complex systems present an ongoing and increasingly important challenge. Over the last few decades computer simulation of interfaces has become vastly more powerful and sophisticated. However, theoretical interface screening remains based on largely heuristic methods and is strongly biased to systems that are amenable to modelling within constrained periodic cell approaches. Here we present an unconstrained and generally applicable non-periodic screening approach for systematic exploration of material's interfaces based on extracting and aligning disks from periodic reference slabs. Our disk interface method directly and accurately describes how interface structure and energetic stability depends on arbitrary relative displacements and twist angles of two interacting surfaces. The resultant detailed and comprehensive energetic stability maps provide a global perspective for understanding and designing interfaces. We confirm the power and utility of our method with respect to the catalytically important $TiO_2$ anatase (101)/(001) and $TiO_2$ anatase (101)/rutile (110) interfaces.

From Bronze Age alloys to advanced ceramics, technological progress heavily relies on the development and use of polycrystalline materials whose performance is largely dictated by crystallite interfaces (i.e. grain boundaries)[1]. Interfaces between different materials are also key to designing heterostructures for use in a range of modern applications (e.g. solar cells[2], photocatalysts[3], quantum dot displays[4]). Here, the formation of well-ordered interfaces is achieved by controlled deposition of one semiconductor on the surface of another which is largely constrained by epitaxial matching[5]. Recently, top-down manipulation of two-dimensional (2D) materials has created a new class of layered heterostructures in which epitaxial constraints are less pronounced due to the relatively weak van der Waals interfacial interactions[6,7]. The resulting freedom to carefully tune the interfaces in such systems (e.g. relative in-plane twist angles of layers[8]) is highly promising for developing the next generation of 2D nanodevices and has already yielded spectacular new phenomena[9]. Although clearly playing a huge role in established and emergent technologies, interfaces are highly complex systems whose properties are typically difficult to predict and/or rationalise. Computational modelling is playing an increasingly important role in helping to analyse and understand interfaces. Recent methodological advances have tended to focus on approaches for searching for detailed low-energy atomic/electronic structures of selected interfaces[10–13]. However, given the huge number of possible ways in which two surfaces can interact, efficient and accurate screening of the energetic/structural landscape of viable interfaces is a pre-requisite for more in-depth investigations.

[1]Dipartimento di Scienza dei Materiali, Università degli Studi di Milano-Bicocca, Via Cozzi 55, 20125 Milano, Italy. [2]Departament de Ciència de Materials i Química Física & Institut de Química Teòrica i Computacional (IQTCUB), Universitat de Barcelona, c/ Martí i Franquès 1-11, 08028 Barcelona, Spain. [3]Institució Catalana de Recerca i Estudis Avançats (ICREA), Passeig Lluis Companys 23, 08010 Barcelona, Spain. ✉e-mail: giovanni.diliberto@unimib.it; s.bromley@ub.edu

Machine learning has been used to screen the structures and energies of metallic tilt grain boundaries, but required prior training with 10,000's of calculated examples[14]. Here we address the screening challenge with a simple, powerful and direct modelling approach which, in principle, allows for rapid, unconstrained and systematic exploration of energies and structures of interfaces between arbitrary solid surfaces.

Theoretical approaches to studying materials interfaces are typically based on first finding low-energy alignments of two infinitely repeating model surfaces. Historically, coincidence site lattice theory (CSLT) and related methods which aim to maximise favourable interfacial atomic overlap have been widely used for this task[15,16]. The use of CSLT-based methods can often provide a useful first step towards developing possible interface models. However, such heuristic geometric approaches have been criticised for not providing a realistic description of the complex array of chemical interactions at interfaces[17]. For the latter, computational modelling employing periodic atomistic and/or electronic structure calculations is often used. Periodic density functional theory (DFT) based calculations, for example, have proven to be extremely useful in providing a deeper understanding of some specific interfaces[18–24]. A drawback of such an approach is that one must define a common crystallographic cell containing both sides of the interface, which necessarily introduces interfacial strain due to the mismatch between the in-plane lattice parameters of each surface[25]. Such calculations are only physically reasonable when the cell-induced strain is relatively small (e.g. <3%), and methods have been developed to find supercells which minimise interfacial strain[26–28]. Interface stabilities can then be estimated by subsequently subtracting strain energies. Structural and physicochemical properties of such inherently mismatched interfaces are, however, less amenable to such post-correction. This overall approach is further limited to a subset of supercells with sizes small enough to comfortably compute using periodic DFT calculations (i.e. less than ~1000 atoms). Constraints due to model-induced strain and practical tractability can thus lead to potentially relevant interfaces being overlooked in a CSLT/strain-dictated screened periodic DFT approach.

Herein, we demonstrate a general non-periodic method to screen the structure and stability of interfaces without the need of heuristically constructing interfaces within the constraints of a common supercell approach. Generally, once two surfaces on either side of an interface plane are defined, four degrees of interfacial freedom remain: the in-plane relative twist angle and in-plane and out-of-plane relative displacements of the two surfaces. From a grain boundary perspective, the definition of the surfaces relative to an interfacial plane (often reduced to a single out-of-plane tilt angle in symmetric systems) together with the in-plane twist angle contribute to the overall misorientation between grains. In-plane and out-of-plane displacements are linked to grain boundary sliding and grain separation, respectively. Within a periodic approach, varying the surfaces constituting a symmetric grain boundary interface is relatively straightforward as the repeat cell is similar for a range of tilt angles. Within our non-periodic disk interface method, the analogous set of interfaces can be modelled, and moreover the tilt angle can be varied more continuously. For sampling in-plane twist angles a periodic approach is relatively more limited due to the above-noted constraints, while our disk-based approach can be applied for arbitrary twist angles. Unlike periodic models, our method can be applied to any two defined surfaces possessing arbitrary lattice parameters and can then provide a systematic survey of atomically accurate interfaces and their relative energetic stabilities for all possible combinations of relative in-plane twists and displacements.

Generally, the disk interface method thus promises to help open the door to more comprehensive and systematic studies of interfaces in many systems (e.g. grain boundaries, heterojunctions). Specifically, the enabled global overview of interfacial possibilities could help in screening for design/optimisation of new interfaces by tuning twist angle/displacement. We confirm the predictive power of our method by applying it to interfaces between surfaces of titania (TiO$_2$). The interaction between surfaces of anatase TiO$_2$ has recently been highlighted in the photoreduction of CO$_2$[29,30]. Here, we focus on the anatase (001)/(101) interface which has been used to help rationalise these experimental findings[31]. By systematically scanning structure versus relative energetic stability for this system, we were able to find a number of new stable interfaces for this system. We also highlight the increased accuracy and detail of our approach with respect to a geometric CSLT-type approach based on atom site overlap. Finally, we apply our method to a more challenging heterojunction between stable surfaces of the distinct anatase and rutile crystal polymorphs of titania. The anatase-rutile interface has attracted huge interest due to its potential importance in helping to explain the efficiency of titania based photocatalysts[32–34].

## Results

The aim of our method is to generate a set of accurate and comparable interface models for any two chosen surfaces in which the twist angle and relative displacement can be arbitrarily and systematically varied. To achieve this, we create models based on two equally sized disk-shaped slabs, each which exhibit one of the two surfaces to form the interface on a circular face. The disks are then brought close together so that the two surfaces on the circular faces directly oppose one another. An interface is then formed by relaxation of the interacting disk system. To construct our disk models, we take a reference system consisting of two independent $x-y$ periodically repeating slabs (i.e. *surf1* and *surf2*) which are each terminated by one of the two surfaces in the $z$-direction. At this point, the structures of the extended surface slabs (i.e. lattice parameters and atomic positions) can be obtained from experimental data or accurate theoretical models. Note that the definition of the reference system does not imply that the *surf1* and *surf2* have a common periodic cell or the same symmetry. A disk is first cut from each surface slab such that the circular area of the respective surfaces on each disk is the same. Generally, each disk can be cut at an arbitrarily chosen displacement from the origin in the $x-y$ plane with respect to in-plane 2D vectors $\mathbf{c}_{surf1}$ and $\mathbf{c}_{surf2}$, respectively. Formally, this is equivalent cutting a disk from *surf1* at the origin and cutting the disk from *surf2* at $\mathbf{s} = (s_x, s_y)$, where $\mathbf{s} = \mathbf{c}_{surf1} - \mathbf{c}_{surf2}$. To model interfaces corresponding to any in-plane shift vector $\mathbf{s}$, we move the $\mathbf{s}$-centred disk by $-\mathbf{s} = (-s_x, -s_y)$ so that the disks have a shared radial axis. Maintaining this common axis, the two disks are then oriented such that the surfaces are parallel and facing one another. In addition, for any choice of $\mathbf{s}$, the disks can be each rotated separately about the axis through their radially aligned centres while keeping the interfacial area constant. As with the displacement, without loss of generality we can assume that one disk is fixed while varying the twist angle, $\alpha$, of the second disk. To avoid spurious structural relaxation on the radially terminated sides of each disk we fix an outer ring of atoms to their positions in the original reference slabs. Although, for any specific calculation, this limits the degree of radial relaxation of each disk as a whole, one can cut disks from reference periodic slabs in which arbitrary constraints have been applied (e.g. isotropic/anisotropic in-plane strain) to take into account long-range structural adaptions of one surface to the other. In the present work we use reference systems based on fully relaxed periodic slabs. A schematic representation of how we form our disk interface models is shown in Fig. 1.

To obtain reasonably accurate and converged structures and relative energetic stabilities of the interfaces, the disks should be both thick enough in the $z$-direction and wide enough in the $x-y$ plane to minimise finite-size effects. The calculated relative energetic stabilities of interfaces formed by any two interacting disks is calculated using the work of separation ($E_{sep}$) – see "Methods". A suitable disk thickness for each surface can be determined from periodic calculations of

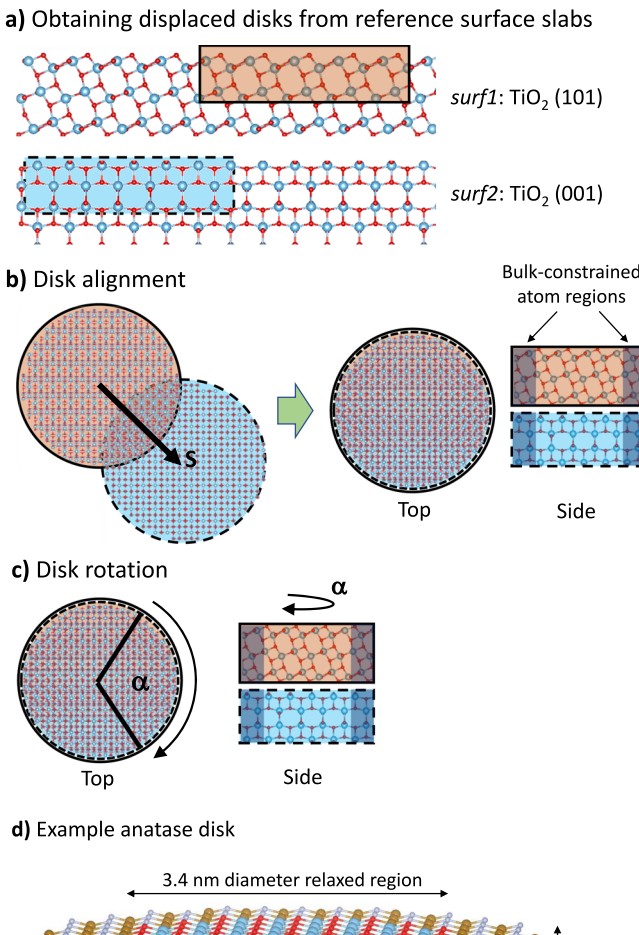

**a) Obtaining displaced disks from reference surface slabs**

*surf1*: TiO$_2$ (101)

*surf2*: TiO$_2$ (001)

**b) Disk alignment**

Bulk-constrained atom regions

Top     Side

**c) Disk rotation**

$\alpha$

Top     Side

**d) Example anatase disk**

3.4 nm diameter relaxed region

1.2 nm

5 nm

**Fig. 1 | Schematic summary of the disk interface approach.** In **a–c**, we show a schematic (not to scale) representation of the process followed to create disk interface models for the TiO$_2$ anatase (101)/(001) interface. **a** Two **s**-displaced disks are cut from the reference TiO$_2$ (101) and (001) slabs. **b** The two disks are translated by **s** and oriented such that the (101) and (001) surfaces face one another. **c** Rotation of the (001) disk by angle $\alpha$ with respect to the (101) disk. Grey shading in the side views indicates regions where atoms are constrained during structure/energy evaluation. In **d**, we show an example of a typical anatase disk. Atom key: unconstrained/constrained Ti atoms – blue/brown, unconstrained/constrained O atoms – red/grey.

converged surface energies. It is important that any finite representation of the surfaces maintains at least these thicknesses throughout the whole interface. Hemispherical cuts from surfaces have been proposed to model energies of grain boundaries in metals with different misorientation angles[35]. The lack of uniform thickness and edge effects in such an approach would likely lead to significant finite effects if applied to more complex inorganic interfaces as studied herein. To avoid such problems very large hemispherical cuts would be required leading to significantly larger and more computationally expensive models relative to a disk approach.

In addition to scanning twist angles, one of the main features of the present disk-based method is the ability to scan relative energetic stabilities for relative displacements at any selected twist angle for two interacting surfaces free from concerns of variable thickness-based

finite-size effects. For an interface between two infinite surfaces, displacing one surface with respect to a fixed second surface and the complementary case are equivalent. For finite systems, this symmetrical relation is not strictly true due to the truncated representation of the infinite translational periodicity of each surface. This finite-size effect can be minimised by using wider disks and its extent for any diameter can be estimated by performing two scans where each disk is displaced using the other disk as a fixed reference. Such complementary scans are used to refine all reported relative energetic stability data in both twist angle and displacement scans (see more details in Supplementary note 1).

By varying the relative in-plane displacement (**s**) and the relative radial twist angle ($\alpha$) of the disks, we can systematically scan arbitrary *surf1*/*surf2* interfaces without constraints of periodicity or symmetry. We note that out-of-plane displacement can also be freely varied in this approach but, for simplicity, in our example case studies the disk-disk separation is fixed in the constrained interfacial regions for each in-plane displacement and twist angle sampled.

**Case study**
To provide a concrete example of how our method works we first consider the TiO$_2$ anatase (101)/(001) interface. To obtain the relaxed structures and relative energetic stabilities of our model interfaces, each corresponding disk interface system is evaluated using computational chemical modelling. Following previously reported periodic DFT modelling studies of the anatase (001)/(101) interface[31] we use reference slabs and disks with ~1.2 nm thicknesses (i.e. six atomic layers). To further ensure reasonably converged energies in this system, we also found that disks of 5 nm diameter were adequate (see Supplementary note 2). Two disks with these dimensions yield (001)/(101) disk interface models with >4000 atoms. To avoid finite-size effects from the disk cutting procedure, the atoms in an outer annulus of each disk were fixed to their bulk crystalline positions (see Supplementary note 3). Considering further that the aim of the approach is to systematically sample a wide range of twist angles and in-plane displacements (each requiring a separate calculation), the direct use of ab initio electronic structure methods such as DFT would currently be computationally prohibitive. Relatively computationally efficient tight-binding electronic structure methods have recently been parameterised for TiO$_2$ and used for calculating interactions between small facetted titania nanoclusters of at least one order of magnitude smaller than our disks[36]. The inter-cluster junctions in such an approach are affected by variable size, thickness and shape with respect to rotation and involve variable edge-surface interactions which preclude them providing reliable models of extended interfaces. Although tight-binding approaches are still somewhat computationally expensive for extensive screening of our large disk interface systems, they could be promising for post analysis of the electronic structure of selected systems where a periodic DFT approach is not viable. For a relatively rapid means to systematically scan in-plane displacements and twist angles for our chosen anatase system, we employ interatomic potentials (IPs). The usefulness and accuracy of computationally efficient empirically parameterised IP for modelling inorganic oxide interface was established in early IP-based periodic calculations[37,38]. Nowadays, as in our study, such IPs are more often used to obtain fairly accurate and efficient results prior to more detailed calculations[10].

We note that machine-learned IPs[39] could also be used for potentially increased accuracy and would also allow general application of our approach to interfaces between materials for which parameterised IPs are not currently available. Herein, we use empirically parameterised IPs that have been confirmed to well describe relative energetics and structures of the bulk, surfaces and nanostructures of TiO$_2$[40–43]. For disks of infinite diameter, the interface would be perfectly periodic and relative shifts of the type **s** = ($s_x \pm A a_1 \pm B a_2$, $s_y \pm$

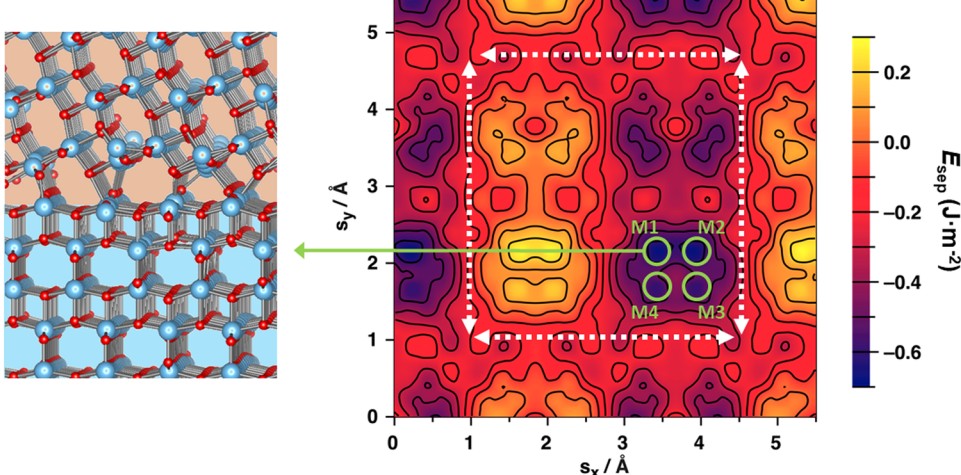

**Fig. 2 | Anatase (101)/(001) relative energetic stability displacement map.** Right: $E_{sep}$ values for $\alpha = 0°$ with respect to changing the $x$ and $y$ components of the shifting vector **s**. The white dashed arrows indicate a repeat region with local translational symmetry according to the relevant in-plane lattice parameters of the two surfaces (see main text). Left: the relaxed atomistic structure of the anatase (101)-upper/(001)-lower interface corresponding to the one of four near degenerate lowest energy minima on the map (M1, M2, M3 and M4 circled in green) is highlighted (see also Supplementary note 5). Atom key: Ti−blue, O−red.

$Cb_1 \pm Db_2$) would yield symmetrically equivalent interface configurations with the same energy, where $a_1$, $a_2$, $b_1$, $b_2$ are the in-plane lattice parameters of *surf1* and *surf2* and A–D are integers (see Supplementary note 4). After compensating for finite-size effects (see Supplementary note 1), the mean absolute deviation between energies of periodically equivalent points was found to be 0.02 J·m$^{-2}$. We note that the adopted disk diameter corresponds to almost five times the largest in-plane lattice parameter, and more than thirteen times the smallest one. Interfaces were modelled with an initial disk-to-disk interfacial separation of ~2.2 Å and the unconstrained atoms in both disks were subsequently fully relaxed (i.e. in $x$, $y$ and $z$ directions) in non-periodic IP-based calculations.

Figure 2 shows the map of relative energetic stabilities ($E_{sep}$) of a different TiO$_2$ anatase (101)/(001) interfaces with respect to systematically varying the shifting vector **s** over a grid of 625 points in the range 0 Å ≤ $s_x$ ≤ 5.6 Å and 0 Å ≤ $s_y$ ≤ 5.6 Å, for a fixed twist angle of $\alpha = 0°$ (i.e. no relative rotation). The map reveals a complex energy landscape with numerous distinct minima and maxima spanning an energy range of the order ~0.8 J·m$^{-2}$. As expected, the translation symmetries of the (001) and (101) surfaces are recovered within the range of shift vectors sampled (see Supplementary note 4). Specifically, the map has translation symmetry in the $y$-direction with a distance close to 3.7 Å. This distance corresponds to the $a$ lattice parameter of both the (001) and (101) anatase surfaces (i.e. 3.69 and 3.79 Å, respectively) within the resolution of our energy sampling (~0.2 Å grid spacing). In the $y$-direction the energy map has a $y$ repeat distance of 3.78 Å corresponding to the $b$ lattice parameter of the anatase (001) surface. The $b$ lattice parameter of the (101) surface is 10.24 Å which is outside of the displacement range sampled. Within this translationally symmetric quasi-square region (see dashed white arrows in Fig. 2) we find four nearly equally sized sub-regions. Two of these sub-regions tend to be energetically more favourable for forming interfaces, whereas the other two mainly correspond to energetically unfavourable interface configurations. Within and around these four main regions there are several minima and maxima which often appear to follow local quasi-regular ordering. Figure 2, for example, highlights the four lowest energy minima (green circles), all with similar energies positioned in quasi-square arrangement. The atomistic structure of the interface corresponding to one of these four nearly degenerate minima is also shown in Fig. 2 (see also Supplementary note 5). Overall, the map reveals an unexpectedly non-trivial and highly detailed dependence of relative interfacial stability on relative translational displacements.

We now investigate the relative energetic stability of interfaces for $\alpha \neq 0$. This corresponds to rotating the (001) disk, with respect to the (101) disk about their common radial axis for each displacement. Figure 3 shows the relative stability displacement maps for $\alpha = 30°$, 45°, and 90°, each based on 625 sampled interfaces. The main overall effect of the twist angle is to cause a global rotation of the $E_{sep}$ displacement map for $\alpha = 0°$. From Fig. 3 it is clearly seen that the relative positions of the maxima/minima in each energy map is very similar, and that they rotate together with respect to the applied twist angle. It is also apparent that the fine detail of the energy landscape of each twisted map is non-uniformly distorted with respect to the $\alpha = 0°$ case. This cursory analysis suggests that varying the shifting vector has a primary effect on determining the relative interface stability, while the contribution from varying the twist angle has a secondary, but still significant, effect.

To determine the angular dependence of the relative stabilities of the interfaces more accurately, we calculated $E_{sep}$ for the four interfaces corresponding to the four lowest energy minima reported in Fig. 2 as a function of $\alpha$. These data, shown in Fig. 3, show a maximum energetic variation of ~0.2 J m$^{-2}$ with respect to changes in twist angle, which is approximately one quarter of the maximum energy variation found with respect to displacements. We note that the angular increment of 5° used in these twist angle scans is of a sufficiently high resolution for capturing the main angular dependent tendencies in relative stability (see Supplementary note 6). Figure 4 clearly shows that the $E_{sep}$ dependence on $\alpha$ is highly complex and is distinct for each of the four lowest energy minima. For example, we find twist angles ranges that are particularly energetically favourable for some minima while being less so for other minima. We note that all four low-energy minima are nearly equally stabilised close to twist angles of $\alpha = \pm 90°$, indicating that, on average, these are particular energetically stable twist angles. Overall, the data summarised in Figs. 2–4 demonstrate the detailed global perspective that can be obtained using our method by directly and systematically sampling the energies of interfaces with respect to **s** and $\alpha$.

Our comprehensive relative energetic stability maps (relative to arbitrary combinations of displacement and twist angle) provide a detailed tool for further interface exploration. As generated, our maps of relative interface energies correspond to locally optimised structures (i.e. allowing for bond formation/rearrangement and atomic relaxation) coming from bringing two surfaces in contact. For interfaces at points of interest on these maps one could then use

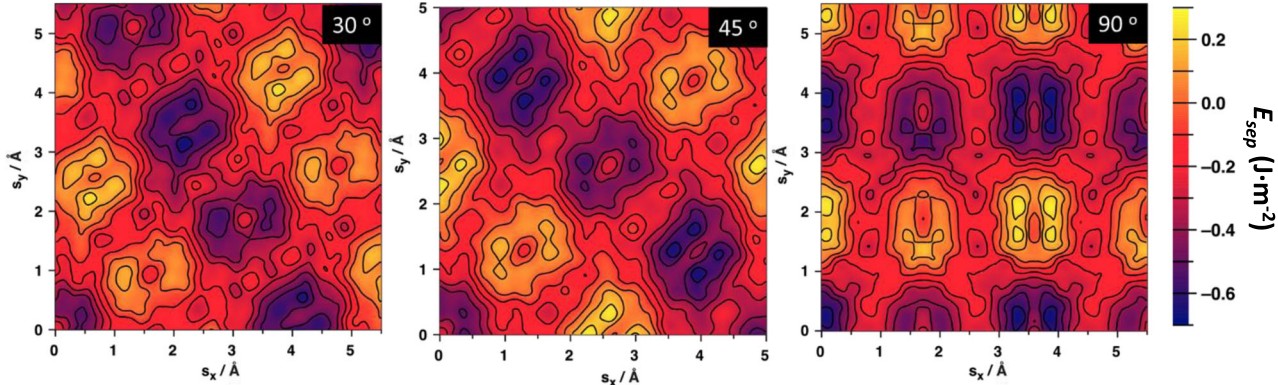

**Fig. 3 | Effect of twist angle on interface displacement maps.** $E_{sep}$ displacement maps for $\alpha = 30°$, $45°$, and $90°$ (from left to right) with respect to the $x$ and $y$ components of the shifting vector **s**.

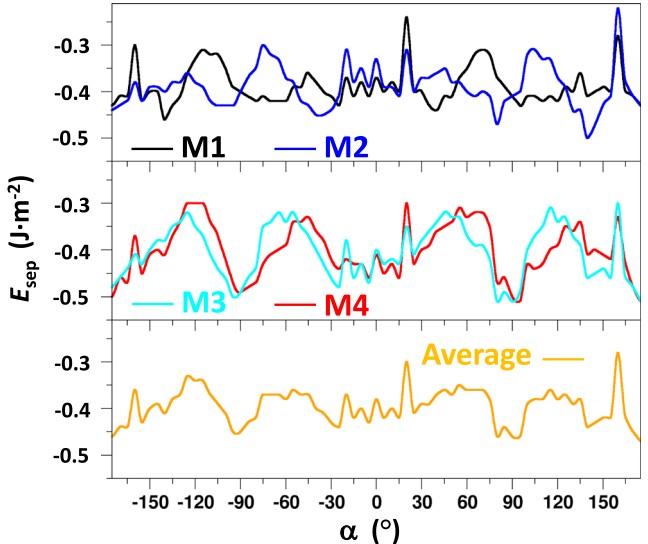

**Fig. 4 | Twist angle dependence of (101)/(001) interface energetic stability.** Calculated $E_{sep}$ values of the four lowest energy (101)/(001) interfaces circled in Fig. 2 (i.e. M1, M2, M3 and M4), as a function of the in-plane twist angle ($\alpha$) sampled every 5°. The lower plot shows the average $E_{sep}$ of all four minima.

constrained molecular dynamics or global optimisation methods to search for more stable reconstructed interface structures[10–13]. The large amount of data in our systematic mappings could also provide a useful set of training data for machine learning approaches to interface screening and analysis[13,14]. As they stand, our predicted interface structures can also be refined with higher-level DFT calculations. In the following section we use periodic DFT calculations to compare the stabilities, structures and properties of the lowest energy anatase (101)/(100) interfaces predicted by the disk interface method with those reported in the literature.

**Model calibration**

To test the accuracy and reliability of our IP-based approach, we performed periodic DFT calculations of the periodic reference slabs of the (101) and (001) surfaces using the hybrid HSE06 density functional (see also Methods). Previous studies have shown that DFT calculations can provide reasonably reliable electronic properties for both anatase surfaces and more demanding extended interface calculations[31,44–47]. Comparing the DFT-optimised reference slabs with the corresponding ones obtained from our IP-based periodic calculations we find a very reasonable agreement between the optimised lattice parameters (see

Supplementary note 7). Following ref. 31, a 2D-periodic anatase (101)/(001) interface model was then built from these slabs in which the (001) surface is twisted by ~45° with respect to the conventional $a$ and $b$ vectors defining the (101) surface plane. This interface was derived by inspection to ensure good cation-anion matching. We note that the optimised periodic supercell containing this interface has lattice vectors $a = 10.923$ Å and $b = 10.527$ Å, which induce moderate lattice strains of ~3%. To assess the consistency of this prediction with our non-periodic IP-based approach, we extracted one of the four near-degenerate lowest energy interface structures (see Fig. 2) predicted from our disk interface displacement scans at $\alpha = 45°$ (see Fig. 3). Using this interface structure as a starting configuration for a periodic DFT geometry optimisation we find that it converges to the same interface structure as that reported in ref. 31. This result confirms the reliability of our proposed approach to provide reliable interface structures, that can be refined at a quantum chemical level of theory without significant change.

To further asses the accuracy of our method with respect to providing accurate relative interfacial energetics, we take the interface structure corresponding to the M1 energy minimum for $\alpha = 90°$ (see Fig. 2). For this twist angle, the DFT-optimised periodic interface has lattice parameters $a = 11.267$ Å and $b = 10.277$ Å, corresponding to an interface strain of ~3%, which is comparable to that for the $\alpha = 45°$ interface for M1. Comparing the DFT-calculated energies for both interfaces, the 90° interface is found to be 0.12 J m$^{-2}$ more stable than the $\alpha = 45°$ interface. This result is in line with that from our IP-based non-periodic disk interface energy scans ($\Delta E = 0.05$ J m$^{-2}$) showing a good agreement between the two approaches. From the IP-based periodic model of our newly predicted $\alpha = 90°$ interface we also calculate its interface energy with respect to bulk anatase (see "Methods") to be 1.02 J m$^{-2}$. These results demonstrate that our method can be used to systematically screen for new stable interfaces. The analysis of the $\alpha = 90°$ interface structure suggests that its relative higher stability is due to a better matching of ions at the interface where nine Ti–O bonds form, as compared with the formation of eight Ti–O bonds when $\alpha = 45°$ (where we take the presence of bond to be indicated by a Ti-O distance in the range 1.9 - 2.2 Å).

Next, we analyse the electronic properties of the DFT-optimised $\alpha = 45°$ and $\alpha = 90°$ interface structures. In particular, we focus on the band edge alignment as this is an important quantity for photocatalytic applications[48,49]. The photoactivity of this system has been linked to a type-II alignment between at this interface, with both valence and conduction band edges in the (001) surface being ~0.5 eV higher in energy than those of the (101) surface[31]. This alignment tends to promote the migration of photo-generated electrons to the (101) side of the interface and holes to the (001) side[29]. We calculated the band offsets of the fully optimised DFT interfaces for both $\alpha = 45°$ and

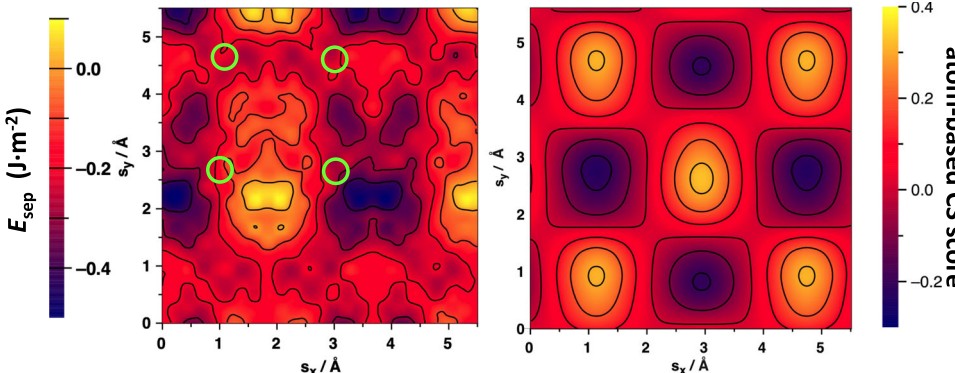

**Fig. 5 | Comparing $E_{sep}$ values with atom-based CS score values.** Comparison between displacement maps for the anatase (101)/(001) interface at $\alpha = 0°$ derived from: (1) explicitly the evaluating the interface structure and relative energy with an IP (left) in our disk interface approach, and (2) by applying a signed atom-based CS measure (right). Green circles highlight the displacements of interfaces with maximum and minimum atomic overlap as predicted by the atom-based CS score. The details of our signed atom-based CS method are described in Supplementary note 9.

$\alpha = 90°$, by means of the potential line-up method[18,50,51]. For both 90° and 45° the $TiO_2$ (101) band edges are lower in energy than those for the (001) surface, leading to a type-II junction (see also Supplementary note 8), in line with what is observed experimentally. Indeed, the (101) band edges are lower by 0.57 eV when $\alpha = 90°$, as compared with 0.41 eV for the $\alpha = 45°$ interface. Apart from this small alignment difference, within the typical error of DFT methods[51,52], both interface models consistently describe the physics behind the $TiO_2$ anatase (101)/(001) surface junction.

In Fig. 5, we compare our results based on explicit energy minimisation of structurally accurate interfaces with those arising from a signed atom-based coincidence site (CS) measure with the rotation angle set at $\alpha = 0°$. Our atom-based CS approach identifies potential interfaces based on maximising the degree of coincidence (or overlap) of atom sites in the two respective surfaces which are treated as rigid geometric lattices (see Supplementary note 9). Our CS method is similar in philosophy to traditional CSLT approaches in that it purely focusses on geometric site coincidence and neglects both the possibility of bond formation/breaking or atomic relaxation at the interface. It is thus expected that its predictions will differ from a more realistic atomistic modelling approach for interfaces with strong interactions. We observe that in both approaches the resulting displacement maps for our anatase-anatase system exhibit the expected overall translational symmetry arising from the unit cell parameters of the respective surfaces. Both maps also show similarly near-equally sized and regularly spaced sub-regions indicating mainly favourable or unfavourable interface configurations. However, in the atom-based CS map these sub-regions contain only one maximum or minimum whereas the interface energy map from our double-disk scan exhibits several minima and maxima. Generally, the atom-based CS map indicates that there are only four distinct displacements of interest for the anatase (101)/(001) interface as compared to the complex energy dependence on displacement predicted by our energy-based maps. Moreover, the four displacements corresponding to maxima/minima of atomic overlap in the atom-based CS map do not match with positions of the lowest (or highest) energy points in the IP-based energy map (see green circled points in Fig. 5). Clearly, our atom-based CS model cannot reproduce the detailed energy landscape derived from an explicit evaluation of relative interface energies. Although this not so unexpected[17], it is perhaps surprising that the qualitative predictions of our atom-based CS model are not even a reasonable guide to find the lowest energy interface in our example. The inadequacies of our atom-based CS model for describing the anatase (101)/(001) interface are even more pronounced for other angles (see Supplementary note 9) which strongly suggests that a more detailed and accurate method is essential to properly explore the complex structural and energetic

space of interface possibilities. The discrepancy between the two approaches is likely due to the numerous strong bonding interactions and local structural distortions occurring at the anatase (101)/(001) interface, that are not accounted for in CS-based approaches. We believe that our disk interface method could be used an efficient and accurate guide to find new interfaces while also providing a systematic and atomistically detailed overview of interface configurations.

Finally, as an example of a heterojunction between two distinct crystal structures, we apply our method to scanning anatase/rutile titania interfaces. The alignment between the anatase and rutile band edges at such interfaces has been invoked to explain the high photo-activity of titania[32,33]. As far as we are aware no detailed models of extended anatase/rutile interfaces has yet been reported and previous modelling has focussed on comparing the properties of separated anatase and rutile systems[34,53]. Here we consider interactions between the most stable anatase (101) and rutile (110) surfaces. Modelling this interface would be very challenging for periodic DFT calculations due to the very high lattice mismatch involved. For the simplest aligned (i.e. $\alpha = 0°$) system, the $a$ and $b$ lattice vectors of anatase (101) are approximately 25% and 60% larger than the corresponding values for rutile (110), implying that large supercells should be needed to minimise the mismatch. A $(4 \times 5)$ anatase (101) supercell could then be used to reasonably match a $(5 \times 8)$ rutile (110) supercell, corresponding to ~1600 atom periodic interface model. This demanding calculation would provide a single interface structure for one displacement and one twist angle. To systematically scan both displacements and twist angles of this system we use a disk interface model based on a rutile disk exhibiting the (110) surface interacting with an anatase disk exhibiting the (101) surface for $\alpha = 0°$ (see "Methods"). Figure 6a shows the resulting $E_{sep}$ displacement map for this system. Clearly the resulting displacement map is more complex than those for the anatase (101)/(001) interface (e.g. see Fig. 2) and shows many distinct maxima and minima. We focus on two of the lowest energy minima (circled in Fig. 6a) and perform an twist angle scan at the corresponding fixed displacements. Figure 6b shows that the most the most energetically favourable angle is $\alpha = 0°$. Finally, we extract the atomistic structure of the most stable interface predicted by these scans (see Fig. 6c) and make a corresponding periodic model following the lattice-matching prescription above. Although such a model is highly demanding to evaluate using DFT, using our computationally efficient IPs we evaluate the corresponding bulk-referenced interface energy to be 0.92 J m⁻². Within the plane of the interface the structure is clearly non-uniform and exhibits areas of high and low-density of anatase-rutile bonding interactions. Of particular interest is the former where the localised strong interaction induces an out-of-plane asymmetric structural distortion extending at least three atomic layers into the

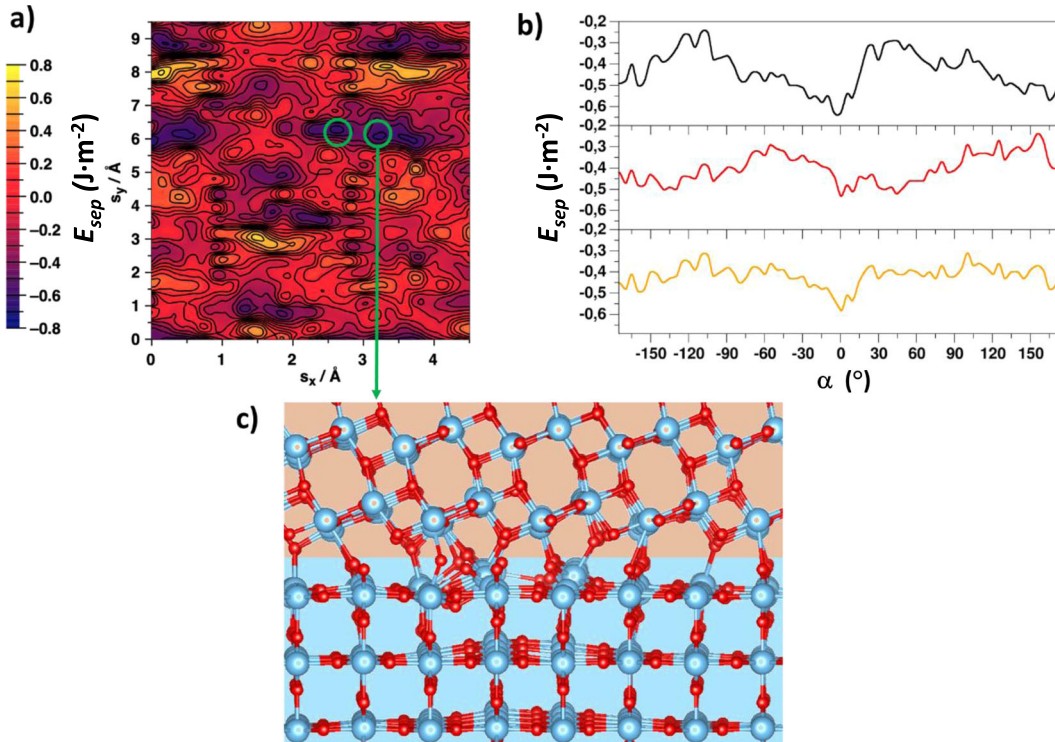

**Fig. 6 | Application of the disk interface method to the anatase (101)/rutile (110) interface. a** Relative energetic stability displacement scan of the anatase (101)/rutile (110) interface at $\alpha = 0°$. **b** Upper two plots (black and red) show twist angle scans for two low-energy displacements circled in **a**. The lower orange line shows the averaged twist angle scan from the two upper plots. **c** The atomistic structure of one of the circled minimum energy anatase (top)−rutile (bottom) interfaces. Atom key: Ti−blue, O−red.

rutile phase, while hardly affecting the structure of the anatase phase. These interesting structural characteristics are likely to have a significant impact on the electronic structure of the interface, with possible consequences for through-interface charge transport and photo-excited hole-electron recombination rates and will be studied in more detail in future work.

In summary, we present a general non-periodic approach for systematic and accurate screening and exploration of materials interfaces. Our disk interface method uses two interacting nano-disks to model the structure and energetics of interfaces between two defined surfaces with arbitrary in-plane relative translational displacements and/or relative twist angles. Scanning over displacements and angles provides a systematic and detailed map of interfacial possibilities that can be interrogated and subsequently refined (e.g. DFT calculations, global optimisation). We first tested our method with respect to the anatase (101)/(001) interface which has relevance to photocatalysis. Detailed systematic scans of thousands of interface structures with respect to displacements and twist angle reveals highly detailed and complex stability maps, which cannot simply be rationalised by geometric atom-based CS considerations. DFT calculations on energetically stable interfaces extracted from these maps, confirms that the disk interface method can both reproduce previously reported interface models and predict new interface structures with relatively higher stabilities. Lastly, we applied our method to the challenging and scientifically/commercially important anatase/rutile $TiO_2$ heterojunction for which we report a new structurally interesting interface that could have important implications for the highly studied photocatalytic properties of this system.

The unconstrained and general nature of the disk interface method means that, in principle, it could be applied to a wide range of disparate interfacial systems. Due to its non-periodic character, the method is particularly well suited to complex non-symmetric interfaces between dissimilar materials and for the inclusion of interfacial trapped charges and defects. Future applications of the disk interface method have the potential to systematically screen as yet unexplored interfaces and provide a fuller and more detailed understanding of the interfacial energy landscapes of important existing systems.

## Methods

We created 1000 s of $TiO_2$ anatase (101)/(001) and anatase (101)/rutile (110) disk interface models each containing >4000 atoms. Interface models were constructed from two face-to-face nanodisks, each with a 5 nm diameter and a 1.2 nm thickness (anatase) or 1.8 nm thickness (rutile) and cut from respective optimised periodic slab calculations. For the present work all surfaces were assumed to be defect-free although we note that the method allows the inclusion of arbitrary defects. Scanning of different interfaces was achieved by choosing disks with differing relative in-plane displacements (i.e. $s_x$ and $s_y$) and by rotating one disk with respect to the other with an in-plane twist angle ($\alpha$)−see also Fig. 1. The optimised structure and relative energy of each model interface was calculated using IP-based[40] partially constrained relaxations employing the GULP[54] code. Specifically, the atoms in the inner part of each disk (3.4 nm diameter) were optimised, while the coordinates of the atoms in outer region were fixed to their original bulk positions in the corresponding reference slabs. The constrained regions of the disks were kept at separated by a fixed interfacial vertical distance of about 2.2 Å during all the scans. Rigid body relaxations of both disks (allowing the disks to move closer or further apart) after the scans resulted in negligible changes in the calculated energies, confirming the adequacy of this choice. Convergence and finite-size testing for the overall nanodisk set-up for the anatase (101)/(001) system can be found in Supplementary notes 1−3.

**Table 1 | Calculated lattice vectors and the number of atoms in the periodic interface slab models used to obtain $E_{int}$ values**

| Model | a/Å | b/Å | Number of atoms |
|---|---|---|---|
| anatase (101)/(001) | 41.262 | 29.605 | 2700 |
| anatase (101)/rutile (110) | 30.057 | 51.317 | 3360 |

Relative interface energetic stabilities for our disk interface systems were calculated using the work of separation ($E_{sep}$):

$$E_{sep} = \frac{E_{disk-interface} - E_{disk1} - E_{disk2}}{2A}$$

where $E_{disk-interface}$, $E_{disk1}$, and $E_{disk2}$ are the total energies of the disk interface system, and the two separate disks respectively, and $A$ is the interfacial area. In this way it is easily possible to identify regions characterised by a favourable interfacing (negative $E_{sep} < 0$) and regions for which interfacing is unlikely ($E_{sep} > 0$). All reported $E_{sep}$ values have been refined by complementary scans (see Supplementary note 1).

In those cases where we extracted a periodic slab model of selected energetically stable interfaces, we also could evaluate the interface energy with respect to the corresponding bulk phases ($E_{int}$):

$$E_{int} = \frac{E_{slab-interface} - nE_{bulk1} - mE_{bulk2}}{2A}$$

where $E_{slab-interface}$ is the total energy of the periodic slab interface model, $E_{bulk1}$ and $E_{bulk2}$ are the bulk energies per formula unit of the two components of the interface, and $n$ and $m$ indicate how many formula units of each bulk system are present in the interface model. $E_{bulk1}$ and $E_{bulk2}$ can be obtained from fully periodic calculations of the respective crystal structures. As $E_{slab-interface}$ is obtained for a periodic slab model which includes two exposed surfaces, their contribution to $E_{int}$ should then be subtracted[24,55]. For our reported $E_{int}$ values, we extracted large periodic slab models from our disk interface scans for a low-energy anatase (101)/(001) interface and a low-energy anatase (101)/rutile (110) interface. Due to the large size of the models and we optimised the structures and obtained all energies using our adopted IPs and the GULP code. The calculated lattice vectors and number of atoms for each periodic interface slab model are reported in Table 1.

Periodic DFT-based calculations using the VASP 6.1.0 code[56–58] were performed to confirm and refine the description of selected low-energy interfaces from the IP-based disk interface scans. Here, the valence electron density was expanded in a plane wave basis set with a kinetic cut-off of 400 eV, where the effect of the core electrons was incorporated by the projector augmented wave approach[59,60]. We employed the Heyd-Scuseria-Ernzerhof (HSE06) range-separated hybrid density functional[61], which has been found to provide a reliable description of the structural and electronic properties of semiconducting metal oxides[62,63]. Bulk anatase was modelled using a single unit cell and a $4 \times 4 \times 4$ Monkhorst−Pack (MP)[64] k-point grid. 2D slab models were used to study the (101) and (001) anatase surfaces (1.2 nm thickness) and the (101)/(001) interfaces (2.6 nm thickness). A $4 \times 4 \times 1$ MP k-point grid was adopted for the slabs, and for the interface models the MP grid was reduced to $1 \times 1 \times 1$ (i.e. gamma point only) because of the relatively large cell dimensions. All slabs were separated from their periodic images in the out-of-plane z-direction by a vacuum space of ~2 nm. All models were structurally optimised until forces on all atoms were less than $10^{-2}$ eV/Å. The periodic models were generated from suitable disk interfaces by: (1) inspecting the nanodisk interface structure, (2) identifying regions of space where atoms are characterised by a near-periodicity, and (3) applying periodic boundary conditions.

## Reporting summary

Further information on research design is available in the Nature Research Reporting Summary linked to this article.

## Data availability

Example results from applying the disk interface model to the anatase (001)/(101) interface are included in Supplementary Information.

## Code availability

The code for preparing the disks from periodic slabs together with instructions on how to use the code are included the Supplementary Information.

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

## Acknowledgements

G.D.L. acknowledges the financial support from the Italian Ministry of University and Research (MIUR) through the PRIN Project 20179337R7. Access to the CINECA supercomputing resources was granted via ISCRAB. G.D.L. gratefully acknowledges the support of Universitat de Barcelona and the computer resources and technical support provided by Barcelona Super Computing Center under the Project HPC-EUROPA3 (HPC17W04A2). We also thank the COST Action 18234 supported by COST (European Cooperation in Science and Technology). S.T.B. and A.M.-G. acknowledge support from the Spanish MICIUN/FEDER RTI2018-095460-B-I00 and María de Maeztu MDM-2017-0767 grants. A.M.-G. also acknowledges support from the Spanish MICIN PID2020-115293RJ-I00/AEI/10.13039/501100011033 grant. The authors thank Profs. Gianfranco Pacchioni and Francesc Illas for useful discussions.

## Author contributions

S.T.B. came up with the original concept. G.D.L. wrote the code used to generate the initial disk interface models. S.T.B. performed the IP-based GULP disk interface calculations. G.D.L. performed the periodic DFT calculations. A.M.-G. produced the relative energy displacement plots. All authors discussed the results and contributed to the manuscript preparation. S.T.B. coordinated the project.

## Competing interests

The authors declare no competing interests.
