## [Peer Review File · Nature Communications]

REVIEWER COMMENTS

Reviewer #1 (Remarks to the Author):

I have read with great interest the manuscript by Di Liberto, Morales-Garcia and Bromley where they propose a new method for solving the challenge of modeling grain boundaries and heterogeneous solid-solid interfaces with atomistic details.

The idea of interacting nano-disks is ingenious and novel, and it will certainly be of great help for the modeling of heterojunction in many fields related to computational materials science.

Overall, the paper is very well written and despite a high degree of technicalities the topic is of great relevance and interest. I only have two major concerns on the manuscript in its current version:

1) I just do not agree when they claim that their system is strain-free, because the edge of the disks are kept frozen to the bulk position. In this case, the softest system is not allowed to adapt to the structure of the hardest one at the interface.

2) The authors tested the new approach on the well-studied interface among two different surface terminations of TiO₂ anatase. Despite the importance of this system, it has been very much explored – also by some of the authors – with standard surface-slab models and periodic DFT calculations. Here in this manuscript, the authors found some new minima, very close to the explored ones, but their energetic, structural and electronic properties are very similar to those already known. No unexpected feature has been found. This is a limit for this manuscript, as it does not present how relevant is the new proposed approach to solve open questions in interfacial chemistry and materials sciences. In other words, there is no conclusive proof that the new approach can lead to a new breakthrough. For such a relevant journal as Nature Communication I would suggest the author to test the proposed approach on a more challenging and new heterogeneous interface, trying their method where periodic slab-based approaches have failed.

Reviewer #2 (Remarks to the Author):

In this study, the structure of TiO₂ anatase (101)/(001) interface is investigated using molecular statics calculations. Typical interface energy calculations often use periodic boundary conditions for computational efficiency and convenience. This approach has limitations because it excludes other boundaries that are not compatible with periodic boundary conditions. The approach presented in

the current work does not rely on periodic boundary conditions, and if I understand correctly the main novelty of the work. Unfortunately, the same approaches have been previously published (Lee B-J, Choi S-H. Modelling Simul. Mater. Sci. Eng. 2004;12:621.) and have been used in many other subsequent studies. This method is not new. Moreover, in their binary system, the method does not attempt to optimize the local chemistry at the interface, which has to be done and has been done in other similar studies. Nor other advanced structure sampling techniques are used or introduced. Therefore, I feel that a publication of the current manuscript in Nature Communications is not justified.

Reviewer #3 (Remarks to the Author):

Comments: the manuscript titled "An Unconstrained Approach to Systematic Structural and Energetic Screening of Materials Interfaces" by Giovanni Di Liberto et al.

In this work, the authors introduced an approach to study the materials interfaces by using non-periodic nanodisk models instead of the traditional periodic supercells. This is actually interesting and important, especially for the asymmetric interfaces and heterostructures. They applied their method to study the anatase TiO₂ (101)/(001) interfaces, and found low-energy structures at certain displacements and twist angles. The manuscript is well written and organized, the method is introduced clearly. Therefore, I would like to recommend it to be published in Nature Communications, but I still have some unclear parts as described below:

(1) The coincidence site lattice theory (CSLT) is the most used method to build initial GB structures since it usually indicates special interfaces. Could the authors explain why the CSL models fail to find the low-energy interfaces in this example?

(2) In the studies of interfaces, removing or adding atoms may lead the interfaces to lower-energy configurations. I am wondering if the authors have considered it, in TiO₂ it is more complicated that the interface stoichiometry may change. Will the energetically stable twist angle and the disk displacement be different?

(3) The interface energies of the minima energy interfaces in this work are negative, which makes the results hard to be compared with other studies of GBs [10][12][13]. This also makes me doubt the formula for calculating interface energies in this work:

$$E_{\text{int}} = (E_{\text{double-disk}} - E_{\text{disk1}} - E_{\text{disk2}})/2A.$$

In principle, the interface energy for a system with two free surfaces is calculated by

$$E_{\text{int}} = (E_{\text{double-disk}} - N \cdot \mu_{\text{TiO}_2})/A - E_{\text{surface1}} - E_{\text{surface2}},$$

where E_{surface1} and E_{surface2} are the free surface energy of disk1 and disk2, N is the total number of TiO₂ pairs in the system, and μ_{TiO_2} is the chemical potential of TiO₂ in bulk.

$$E_{\text{surface}} = (E_{\text{disk}} - n \cdot \mu_{\text{TiO}_2})/2A,$$

n is the number of TiO₂ pairs in one nanodisk. In the end, the interface energy is

$$E_{\text{int}} = (2E_{\text{double-disk}} - E_{\text{disk1}} - E_{\text{disk2}} - N \cdot \mu_{\text{TiO}_2}) / 2A.$$

Although the authors focused on the relative stability between different interfaces, negative interface energies could mislead readers to understand the stability of obtained structures and decrease the accuracy of this work.

(4) What is the increment during the rotation of one disk?

(5) On page 9, the authors wrote "Specifically, the map has translation symmetry in the y-direction with a distance close to 3.7 Å ..." I think it is x-direction instead of "y-direction", please correct it.

(6) In the DFT calculation part, the authors extracted structure from the lowest-energy interface, I am wondering how it was done since the interface has a large area and is non-periodic.

RESPONSE TO REVIEWERS

Reviewer 1

I have read with great interest the manuscript by Di Liberto, Morales-Garcia and Bromley where they propose a new method for solving the challenge of modeling grain boundaries and heterogeneous solid-solid interfaces with atomistic details. The idea of interacting nano-disks is ingenious and novel, and it will certainly be of great help for the modeling of heterojunction in many fields related to computational materials science. Overall, the paper is very well written and despite a high degree of technicalities the topic is of great relevance and interest. I only have two major concerns on the manuscript in its current version:

1. I just do not agree when they claim that their system is strain-free, because the edge of the disks are kept frozen to the bulk position. In this case, the softest system is not allowed to adapt to the structure of the hardest one at the interface.

Our claim to being strain-free is made with respect to the artificial surface1/surface2 strains due to in-plane mismatches induced in periodic supercell approaches. This is now more clearly stated. The reviewer is correct that by fixing of the edges of the disks to the optimized bulk crystalline structure our method restrains the full in-plane relaxation of the system for any single calculation. However, for any interface one can employ disks that have been cut from slabs that have been optimized with arbitrary constraints (e.g. isotropic or anisotropic in-plane compression/tensions). By doing so one can sample interfaces in which either surface structure is able to better adapt to the other. Following the reviewer's point, a series of displacement scans in which the less rigid surface disk is cut from slabs which have been optimised with slight in-plane compression/tension could provide an even more systematic pre-screening of interface energies before specific angular scans. We thank the reviewer for raising this issue which we now briefly discuss in the revised manuscript.

2. The authors tested the new approach on the well-studied interface among two different surface terminations of TiO₂ anatase. Despite the importance of this system, it has been very much explored – also by some of the authors – with standard surface-slab models and periodic DFT calculations. Here in this manuscript, the authors found some new minima, very close to the explored ones, but their energetic, structural and electronic properties are very similar to those already known. No unexpected feature has been found. This is a limit for this manuscript, as it does not present how relevant is the new proposed approach to solve open questions in interfacial chemistry and materials sciences. In other words, there is no conclusive proof that the new approach can lead to a new breakthrough. For such a relevant journal as Nature Communication I would suggest the author to test the proposed approach on a more challenging and new heterogeneous interface, trying their method where periodic slab-based approaches have failed.

Although we feel that the detailed description our new approach and the confirmation that it works should itself be of high relevance and general interest, we accept the reviewer's point. As an example of a challenging new heterogeneous interface, we have selected the TiO₂ anatase-rutile system. Specifically, we examine the interface between the most stable (101) surface of anatase the most stable (110) surface of rutile. The motivation behind selecting this interface is manifold:

1. It is an example of heterojunction between two different crystal phases – unlike our original

anatase-anatase example.

2. This system has attracted a huge amount of attention over the past few years due to its potential role in enhancing the photoactivity of commercial titania photocatalysts (e.g. Degussa/Evonik P25). Selected new references are cited in the revised manuscript.

3. This interface is very technically challenging due to the very high surface lattice mismatch. Specifically, the a lattice vector of anatase is 25% larger than that of rutile, and the corresponding b vector of anatase is 60% larger than its counterpart. For a periodic calculation at zero twist angle could use a (4x5) anatase (101) supercell with a (5x8) rutile (110) supercell, to help minimize the interfacial strain. This would require system size of about 1600 atoms making it very computationally expensive for a DFT calculation. This calculation, to the best of our knowledge, has never been reported. However, the main concern would be that this expensive calculation would represent only a single interface structure with one displacement and twist angle, and other candidate interfaces would be highly non-trivial to sample. We now report a systematic scan of this system and thus report the first candidate structure of a low energy extended anatase (101) – rutile (110) interface.

4. Considering the constraints of article length, this system allows to naturally extend our study of the anatase/anatase system while retaining a similar computational set-up (e.g. the same reliable interatomic potentials). Although, we do not have space to provide an in-depth study of this new system we do note that the low energy interface structure we find is non-trivial in a number of respects which may be linked to the transfer of photo-generated charges envisioned at this interface. This new result will form the basis of more detailed studies in the future.

Reviewer 2

Referee wrote: In this study, the structure of TiO₂ anatase (101)/(001) interface is investigated using molecular

statics calculations. Typical interface energy calculations often use periodic boundary conditions for computational efficiency and convenience. This approach has limitations because it excludes other boundaries that are not compatible with periodic boundary conditions. The approach presented in the current work does not rely on periodic boundary conditions, and if I understand correctly the main novelty of the work. Unfortunately, the same approaches have been previously published (Lee B-J, Choi S-H. Modelling Simul. Mater. Sci. Eng. 2004;12:621.) and have been used in many other subsequent studies. This method is not new.

We thank the reviewer for pointing out the work of Lee and Choi, which we cite in the revised manuscript. We note that our work quite distinct to this previous approach with respect to a range of technical and methodological, and application considerations which we outline below:

1. Technical: Our interface models are based on disks of constant thickness which are cut from converged periodic slab calculations. Our approach can thus take advantage of the standard and well-established periodic approach to model surfaces for reference systems, knowing that the thickness is converged throughout each cut disk. The radially symmetric vertical cut used to obtain the disks and the subsequent freezing of the edge regions of the disks to the original slab structure also ensures a uniform and minimally disruptive embedding environment. The work of Lee and Choi use hemispherical cuts which have non-uniform thickness and a complex curved cut through the original material. This approach is highly likely to lead to less cleanly converged systems with more possibilities for finite size effects. We also note that, for a given surface area, disk cuts also lead to systems with fewer atoms, thus making our disk-based approach more computationally efficient.

2. Methodological: We use our method to scan both relative in-plane displacements and scan twist angles between two interacting surfaces. The displacement scans are fundamental and give rise to the most significant variations in interface energies. Once a displacement scan has been performed, we can then refine our scans with subsequent twist angle scans. As far as we are aware, the approach of Lee and Choi is designed for twist angle scans and has not been extended to displacement scans. We note that having cuts from disks with uniform thickness are better suited for scanning in-plane displacements.

3. Application: We highlight our method with respect to an important inorganic system in which we consider interfaces made from: i) different surfaces of the same crystal structure, and ii) a heterojunction based on different surfaces of two different crystal structures. Our method is closely tied to the established world of periodic slab modelling of surfaces and, in principle, can be applied to any system for which such modelling is tractable (e.g. inorganic, organic, metals, mixed-junctions, etc). The approach of Lee and Choi is designed for grain boundaries in metals. As far as we are aware it has not been applied in a more general context. We also note that hemispherical cuts would be considerably more disruptive for materials with more open directionally-bonded structures as opposed to closed packed isotropically bonded metals.

Moreover, in their binary system, the method does not attempt to optimize the local chemistry at the interface, which has to be done and has been done in other similar studies. Nor other advanced structure sampling techniques are used or introduced.

For every interface in our displacement and twist angle scans the local chemistry is optimized in the relaxed regions of our disks (and within the disks). This means that the creation and/or breaking of bonds and local structural distortions (e.g. bond lengths/angles) are fully taken into account at each modelled interface. We note in the manuscript that these locally optimized interfaces could also be subsequently studied in more detail using more advanced structure sampling (e.g. molecular dynamics, global optimization methods) which will be examined in future work.

Reviewer 3

In this work, the authors introduced an approach to study the materials interfaces by using non-periodic nanodisk models instead of the traditional periodic supercells. This is actually interesting and important, especially for the asymmetric interfaces and heterostructures. They applied their method to study the anatase TiO₂ (101)/(001) interfaces, and found low-energy structures at certain displacements and twist angles. The manuscript is well written and organized, the method is introduced clearly. Therefore, I would like to recommend it to be published in Nature Communications, but I still have some unclear parts as described below:

1. The coincidence site lattice theory (CSLT) is the most used method to build initial GB structures since it usually indicates special interfaces. Could the authors explain why the CSL models fail to find the low-energy interfaces in this example?

CSLT uses the structural information of two separated surfaces of the interface, but it neglects their atomistic nature and thus formation/breaking/modification of chemical bonds and local structural distortions at the interface are not considered. Our interfaces have regions of strong interaction in which we have multiple bond formation and local structural distortions which appears

to preclude the use of CSLT.

2. In the studies of interfaces, removing or adding atoms may lead the interfaces to lower-energy configurations. I am wondering if the authors have considered it, in TiO₂ it is more complicated that the interface stoichiometry may change. Will the energetically stable twist angle and the disk displacement be different? In the present work we focus on stoichiometric defect-free surfaces for the purpose of demonstrating our method on well-defined clean systems. The reviewer raises the interesting and important issue of defects which indeed will be present to some degree in any real interface. Defects at surfaces are, unfortunately, very rarely considered in traditional DFT periodic modelling of interfaces – largely due to technical issues and computational cost. However, our non-periodic double-disk approach potentially provides a perfect platform for investigating the energetic/structural consequences of defects at surfaces which we will consider in future studies.

3. The interface energies of the minima energy interfaces in this work are negative, which makes the results hard to be compared with other studies of GBs [10][12][13]. This also makes me doubt the formula for calculating interface energies in this work: $E_{int}=(E_{double-disk}-E_{disk1}-E_{disk2})/2A$. In principle, the interface energy for a system with two free surfaces is calculated by $E_{int}=(E_{double-disk}-N*\mu_{TiO2})/A - E_{surface1} - E_{surface2}$, where $E_{surface1}$ and $E_{surface2}$ are the free surface energy of disk1 and disk2, N is the total number of TiO₂ pairs in the system, and μ_{TiO2} is the chemical potential of TiO₂ in bulk. $E_{surface}=(E_{disk}-n*\mu_{TiO2})/2A$, n is the number of TiO₂ pairs in one nanodisk. In the end, the interface energy is $E_{int}=(2E_{double-disk}-E_{disk1}-E_{disk2}-N*\mu_{TiO2})/2A$. Although the authors focused on the relative stability between different interfaces, negative interface energies could mislead readers to understand the stability of obtained structures and decrease the accuracy of this work.

The reviewer is correct that interface energies are typically positive quantities. In our work we follow the common practice the periodic modelling community of quoting the energy gained upon forming an interface which, formally speaking should be termed the work of separation. We also note that in the case of stoichiometric systems (as in our case) the chemical potential term in the energy expression can be removed. This issue is concisely covered in section 3.1 of ref. 24 (i.e. J. Phys.: Energy 1 (2019) 016001). We now clearly state that more negative values of our “interface energy” should be considered as more stable.

4. What is the increment during the rotation of one disk?

The increment in our angular scans is 5 degrees. This is found to be a reasonable compromise between computational cost and accuracy of the sampling. This is now noted.

5. On page 9, the authors wrote "Specifically, the map has translation symmetry in the y-direction with a distance close to 3.7 Å ..." I think it is x-direction instead of "y-direction", please correct it.

This is now corrected.

6. In the DFT calculation part, the authors extracted structure from the lowest-energy interface, I am wondering how it was done since the interface has a large area and is non-periodic.

When extracting an initial interface structure from a double-disk model, one passes from a non-periodic system to a periodic system. Therefore, we simply looked for regions in the disk interface which showed a near-periodicity, extracted the atomic coordinates, and placed them in a suitably sized unit cell to create a periodic model. In the revised manuscript we better explain this procedure.

REVIEWER COMMENTS

Reviewer #1 (Remarks to the Author):

The authors have successfully addressed all my former concerns on the work. The addition of a new less explored interface is very relevant. Overall the manuscript is sensibly improved and I can now recommend publication in its present form.

Reviewer #3 (Remarks to the Author):

The revised manuscript is slightly improved, but there are still problems.

The in- and out-plane displacements of the two grains are important for finding low-energy interfaces, but they are usually considered in the present global structure prediction methods. The more interesting thing is the twist angle, which is also the main point the authors want to present in this manuscript. Unfortunately, the authors did not find any special angles rather than the angles of 0, 45, and 90 degrees which are conventionally considered in interface studies. And the structure with a 45-degree twist angle, according to the authors, is equivalent to the structure in the authors' previous work [30] which has been already discussed thoroughly.

The results of anatase (101)/(001) interfaces show clearly that the low-energy interfaces appear with different in-plane displacements, but in the result of anatase (101)/rutile (110) interface, the authors rotated the disks with fixed displacements and concluded the most energetically favorable angle is 0 degree, I do not think this result is convincing. Also, in their cited article Ref[35], the anatase (101)/ rutile (110) interface has been discussed and the twist angle of the most stable interface is around 30 degrees, but the authors did not make any comparisons and explain the differences.

The authors try to show their disk models can find lower-energy interfaces than the CSL models, but I do not think the comparison is reasonable. The CSL models give interfaces with special symmetries, and they are sensitive to the twist angles, the CSLs can be formed only at certain twist angles. The authors did not present the possible low-energy GBs by CSLT correctly, for example, how large the twist angles are and the corresponding Sigma. Simply using surface atomic positions overlap is not a proper way to describe the CSL models. The increment for angle scanning is 5 degrees, which could also lead to certain CSL GBs missing.

As I wrote in the previous review comments, for finding lower energy interfaces, it is fine to use the formula, the work of separation, to compare the relative stability, but it is not correct to call it "interface energy" directly. The chemical potential terms are canceled out only in a periodic system without free surfaces, which is clearly stated in the paper the authors presented, J. Phys.: Energy 1 (2019) 016001. I do not expect the authors to calculate the accurate interface energy for all interfaces, but at least for the obtained minimum energy structures, to show how stable these interfaces actually are and make them comparable with other works.

I do not agree with what the authors wrote, defects at surfaces were very rarely considered in traditional DFT periodic modeling of interfaces. Adjusting the interface atomic density is widely used in modern structure predictions, for example in these works PRL 96, 055505 (2006), Nature Materials 9, 418–422 (2010), Nat. Commun. 4, 1899 (2013), Nat. Commun. 12, 811 (2021).

Overall, there is no significant breakthrough in the results, and considering the different aspects shown above, I do not think this manuscript is proper to be published in Nature Communications.

REVIEWER COMMENTS

Reviewer 1

The authors have successfully addressed all my former concerns on the work. The addition of a new less explored interface is very relevant. Overall the manuscript is sensibly improved and I can now recommend publication in its present form.

We thank the reviewer for his/her positive assessment of our revised manuscript.

Reviewer 3

The revised manuscript is slightly improved, but there are still problems.

We are pleased that the reviewer appreciates our significant efforts to improve the manuscript. We address the remaining comments/clarifications below.

1. The in- and out-plane displacements of the two grains are important for finding low-energy interfaces, but they are usually considered in the present global structure prediction methods. The more interesting thing is the twist angle, which is also the main point the authors want to present in this manuscript. Unfortunately, the authors did not find any special angles rather than the angles of 0, 45, and 90 degrees which are conventionally considered in interface studies. And the structure with a 45-degree twist angle, according to the authors, is equivalent to the structure in the authors' previous work [30] which has been already discussed thoroughly.

As far as we are aware, global structure prediction models using periodic models can only consider displacements at certain twist angles for which supercells can accommodate both surfaces of the interface with a small lattice mismatch. The main novelty of our study is the development of a new approach to efficiently screen interfaces where we can **scan arbitrary combinations of displacement and twist angle**. We chose the anatase (101)/(001) interface as a solid benchmark (i.e. the results of ref [30])

so that we could verify that our method can find stable interfaces (i.e. for 45°). Here, we also find a new low energy interface angle (i.e. 90°) that is more stable than the 45° interface and was not previously reported, demonstrating the predictive power of our method. The values of the twist angles emerging from our study are of secondary importance to the demonstration of the utility/power of the method. The reviewer states that twist angles of 0°, 45° and 90° are “conventionally considered in interface studies”. For periodic computational modelling of interfaces, this is only possible if the cell parameters of both surfaces allow moderately sized low-strain interface supercells to be built for these angles – which is generally not the case.

2. The results of anatase (101)/(001) interfaces show clearly that the low-energy interfaces appear with different in-plane displacements, but in the result of anatase (101)/rutile (110) interface, the authors rotated the disks with fixed displacements and concluded the most energetically favorable angle is 0 degree, I do not think this result is convincing. Also, in their cited article Ref[35], the anatase (101)/ rutile (110) interface has been discussed and the twist angle of the most stable interface is around 30 degrees, but the authors did not make any comparisons and explain the differences.

The newly added study of the anatase (101)/rutile (110) interface follows our experience with the anatase (101)/(001) interface. For the latter system we established that the relative interface energy variance relative to displacement is significantly larger than for rotations. Displacement scans at different twist angles act mainly to rotate the resultant energy map with less significant angular induced energy variations. As such, a single displacement scan is sufficient to find the optimal displacement, and then at this displacement a systematic twist angle scan can be used to fine tune the interface. This is the procedure followed for the anatase (101)/rutile (110) interface. It is unclear why the reviewer finds our results for this new interface unconvincing, but we are happy to include further data if deemed necessary.

The study in ref. 35 (now 36) uses small finite faceted titania clusters (at least one order of magnitude smaller than our disks) and joins them together in various ways. The small inter-cluster junctions in this study (of the order one surface unit cell) will incur significant finite size effects (see our SI for details of such effects and how we deal with them). Further, the inter-cluster junctions have variable size and shape with respect to rotation and involve variable edge-surface interactions, all due to the faceted nature of the clusters used. Due to these issues, the cluster-junction models are not good representations of extended interfaces. Our general method avoids the problems encountered in ref. 35 by using large circular disks in which the interface area/shape is constant with rotation. These differences likely account for the different predictions regarding twist angle. An extra sentence explaining the differences between our method and that of ref. 35 has been added to the revised manuscript.

3. The authors try to show their disk models can find lower-energy interfaces than the CSL models, but I do not think the comparison is reasonable. The CSL models give interfaces with special symmetries, and they are sensitive to the twist angles, the CSLs can be formed only at certain twist angles. The authors did not present the possible low-energy GBs by CSLT correctly, for example, how large the twist angles are and the corresponding Sigma. Simply using surface atomic positions overlap is not a proper way to describe the CSL models. The increment for angle scanning is 5 degrees, which could also lead to certain CSL GBs missing.

The CSL approach is traditionally applied to grain boundaries (GBs) of densely packed mono-elemental systems (typically metals). For a pure twist GB one can first align the lattice parameters of the two grains and then twist one lattice with respect to the other until one finds specific angles with periodic lattice site coincidences. Although CSL does not consider atoms, the idea is based on finding GBs with a good atomic fit and are which are thus expected to have higher energetic stabilities. CSL has often proven to be a useful geometric guide to low energy metal GBs.

In our work, we directly evaluate the relative energetic stabilities of interfaces between dissimilar surfaces of a semi-ionic compound in a non-periodic system (i.e. not a typical GB system). The idea was to extract a CSL-like measure from our system to compare with our directly calculated energetic stability.

Simple geometric measures such as CSL for understanding interface stability have been reported as being lacking with respect to more direct atomistic models (e.g. “It is concluded that no general and useful criterion for low energy can be enshrined in a simple geometric framework. Any understanding of the variations of interfacial energy must take account of the atomic structure and the details of the bonding at the interface.” – A. P. Sutton, R. W. Balluffi (1987), *Overview no. 61: On geometric criteria for low interfacial energy*, *Acta Metallurgica*, 35, 2177). In our study we wanted to use a CSL-type measure that was well tailored for our systems and thus less susceptible to such criticism. In particular, the relatively high complexity of our systems means that a traditional CSL approach has limited applicability:

1.) When considering heterojunctions, traditional CSL does not give a unique prescription for finding the relative lattice displacement from where to start twisting. Therefore, we first perform a scan of rigid body displacements to obtain a low energy starting point for our twist angle scans. For consistency we use these displacements for both our coincidence site scans and our energy evaluations with respect to twist angle. Note that these displacements are determined independently of the underlying lattice sites for the respective surfaces.

2.) Traditional CSL is purely geometric and does not consider atoms/ions and their interactions. Thus, coincidence site lattices for atomically heterogeneous systems can yield highly repulsive and energetically unstable interfaces. To take into account such cases, we give each atomic site a sign and a local spatial region (Gaussian distribution) and calculate the overlap between cation/anion sites in our coincidence measure. This means that we go beyond a purely lattice based approach to a measure which uses the lattices of atoms positions and their degree of coincidence. We note that even some of the very early coincidence site studies used atoms positions to rationalise observed interfaces: Kronberg, M. L. and F. H. Wilson (1949), “Secondary recrystallization in copper”, *Trans. Met. Soc. AIME*, 185, 501 (>750 citations). As such, our coincidence/overlap site measure is not directly related to the number of geometric lattice coincidence sites (and thus it is not characterised by sigma values) as in traditional CSL. We understand that referring to our approach as “CSL” may have caused some confusion and we are happy to use an alternative name: (e.g. “Atom-based Coincidence Site” approach.). In the revised SI we now detail why we chose an atom-based CS method as opposed to a more traditional CSL measure.

As this comparison is not a core part of our study, we originally compared the predictions of our atom-based CS model with our explicit double disk approach for a selected set of example angles corresponding to low energy interfaces. As suggested by the reviewer, we now show the predictions of our model for a systematic wide scan of angles with a significantly smaller increment (i.e. 1°). The new scan is consistent with our original scan, showing that it is highly unlikely that we missed any special angles/symmetries.

4. As I wrote in the previous review comments, for finding lower energy interfaces, it is fine to use the formula, the work of separation, to compare the relative stability, but it is not correct to call it “interface energy” directly. The chemical potential terms are canceled out only in a periodic system without free surfaces, which is clearly stated in the paper the authors presented, *J. Phys.: Energy* 1 (2019) 016001. I do not expect the authors to calculate the accurate interface energy for all interfaces, but at least for the obtained minimum energy structures, to show how stable these interfaces actually are and make them comparable with other works.

As previously mentioned, we follow the common practice the periodic modelling community in the use of the work of separation (E_{sep}) to compare interface stabilities. We appreciate the reviewer’s concern that directly referring to these values as “interface energies” could lead to some confusion. We now directly refer to our calculated E_{sep} values as providing relative energetic stabilities of interfaces. To obtain the requested interface energies relative to the corresponding bulk phases we extracted interface structures from our disk scans to make large periodic extended interface models. The resulting energies of the periodic models of the interfaces, separate slabs and bulk phases were then used to calculate bulk-

referenced interface energies. We find that most stable anatase (101)/(001) interface has an energy of $1.02 \text{ J}\cdot\text{m}^{-2}$ and the most stable anatase (101)/rutile (110) interface has an energy of $0.92 \text{ J}\cdot\text{m}^{-2}$.

5. I do not agree with what the authors wrote, defects at surfaces were very rarely considered in traditional DFT periodic modeling of interfaces. Adjusting the interface atomic density is widely used in modern structure predictions, for example in these works PRL 96, 055505 (2006), Nature Materials 9, 418–422 (2010), Nat. Commun. 4, 1899 (2013), Nat. Commun. 12, 811 (2021).

Although studies on interface structure prediction have considered reconstructed interfaces and/or defects, we refer to the large body of “traditional DFT periodic modelling of interfaces” which focus on OK modelling of interfaces by lattice matching supercell DFT calculations. These studies rarely consider defects at interfaces. Of the important works cited by the reviewer the Nat. Commun. (2021) and the Nat. Mater. (2010) paper focus on new methods for searching for reconstructed interface structures and DFT is used only as final stage refinement of their search results (as in our study). The PRL (2006) and the Nat. Commun. (2013) are more interested in defects but only use interatomic potentials and not DFT. None of these studies are focused on “traditional DFT modelling of interfaces”.

6. Overall, there is no significant breakthrough in the results, and considering the different aspects shown above, I do not think this manuscript is proper to be published in Nature Communications.

As recognised in the reviewer’s first mainly positive report, our main breakthrough is the development, demonstrated validity and predictive power of “...an approach to study the materials interfaces by using non-periodic nanodisk models instead of the traditional periodic supercells. This is actually interesting and important, especially for the asymmetric interfaces and heterostructures.” Moreover, the revised manuscript now contains a new breakthrough result with a prediction of a new stable anatase (101)/rutile (110) interface. This complex and highly important interface has been the focus of a huge amount of scientific/commercial work in photocatalysis but until now there has never been an accompanying detailed structural model.

Of the above points made by reviewer, 1 and 2 focus on the novelty of the specific results coming from applying our new method. While we do not agree with the comments, these points do not question the novelty of the method itself, which is our main result. Point 5 is disagrees with a comment we made in our previous response, which we clarify. Points 3 and 4 relate to concerns raised in the original report. Clearly, the reviewer feels that we did not address these points adequately in our response. Point 3 concerns our coincidence site (CS) measure. Although this is not a core part of our study, we feel that we have provided a valid and instructive comparison between our method and a geometric atom-based CS approach which is tailored for our systems. Point 4 concerns the use of the term “interface energy”, which we have now clarified to avoid any potential confusion. As requested, we also now calculate the bulk-referenced interface energies for our most stable interfaces for each system studied.

The reviewer’s first report recommended publication and raised a few points (mostly minor corrections/clarifications). We attempted to address all these concerns concisely and respectfully in our initial response. We now provide more detail in our responses above and revisions to the manuscript and SI. The reviewer also notes that the revised manuscript is slightly improved. Considering that the latest points raised do not impinge upon our main results and do not require any major revision we are at a loss to understand this final comment in the reviewer’s latest report. We hope that our responses are now sufficiently detailed and convincing to address all the reviewer’s specific concerns.

REVIEWERS' COMMENTS

Reviewer #1 (Remarks to the Author):

I confirm I find this work nice and neat. The revised manuscript clarifies even more the technical novelties introduced in this work and the new insights on the chosen case studies on TiO₂-based interfaces. While I am not particularly fond of the choice of these cases studies based on the same material, they succeed in showing the potentialities of this new approach and thus I think the manuscript deserve publication in this or other high impact scientific journal.